# Trap and Replace: Defending Backdoor Attacks by Trapping Them into an Easy-to-Replace Subnetwork

**Haotao Wang**
University of Texas at Austin
htwang@utexas.edu

**Junyuan Hong**
Michigan State University
hongju12@msu.edu

**Aston Zhang**
Amazon Web Services
astonz@amazon.com

**Jiayu Zhou**
Michigan State University
jiayuz@msu.edu

**Zhangyang Wang**
University of Texas at Austin
atlaswang@utexas.edu

## Abstract

Deep neural networks (DNNs) are vulnerable to backdoor attacks. Previous works have shown it extremely challenging to unlearn the undesired backdoor behavior from the network, since the entire network can be affected by the backdoor samples. In this paper, we propose a brand-new backdoor defense strategy, which makes it much easier to remove the harmful influence of backdoor samples from the model. Our defense strategy, *Trap and Replace*, consists of two stages. In the first stage, we bait and trap the backdoors in a small and easy-to-replace subnetwork. Specifically, we add an auxiliary image reconstruction head on top of the stem network shared with a light-weighted classification head. The intuition is that the auxiliary image reconstruction task encourages the stem network to keep sufficient low-level visual features that are hard to learn but semantically correct, instead of overfitting to the easy-to-learn but semantically incorrect backdoor correlations. As a result, when trained on backdoored datasets, the backdoors are easily baited towards the unprotected classification head, since it is much more vulnerable than the shared stem, leaving the stem network hardly poisoned. In the second stage, we replace the poisoned light-weighted classification head with an untainted one, by re-training it from scratch only on a small holdout dataset with clean samples, while fixing the stem network. As a result, both the stem and the classification head in the final network are hardly affected by backdoor training samples. We evaluate our method against ten different backdoor attacks. Our method outperforms previous state-of-the-art methods by up to 20.57%, 9.80%, and 13.72% attack success rate and on-average 3.14%, 1.80%, and 1.21% clean classification accuracy on CIFAR10, GTSRB, and ImageNet-12, respectively. Code is available at https://github.com/VITA-Group/Trap-and-Replace-Backdoor-Defense.

## 1 Introduction

Deep neural networks (DNNs) have been successfully used in many high-stakes applications such as autonomous driving and speech recognition authorization. However, the data used to train those systems are often collected from potentially insecure and unknown sources (e.g., crawled from the Internet or directly collected from end-users) [1, 2]. Such insecure data collection process opens the door for backdoor attackers to upload and distribute harmful training samples that can secretly inject malicious behaviors into the DNNs (e.g., recognizing a stop sign as a speed-limit sign). More specifically, backdoor attacks add premeditated backdoor triggers (e.g., a tiny square pattern or an invisible additive noise) to a small portion of training samples with the same target label. Such

36th Conference on Neural Information Processing Systems (NeurIPS 2022).

**Stage 1: Bait and trap the backdoor into the classification head.**

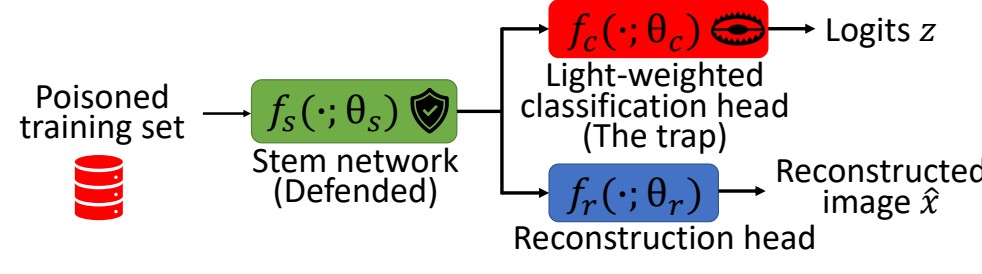

- - - - - - - - - - - - - - - - - - - - - - - - - - - - - - - - - - -

**Stage 2: Replace the infected classification head.**

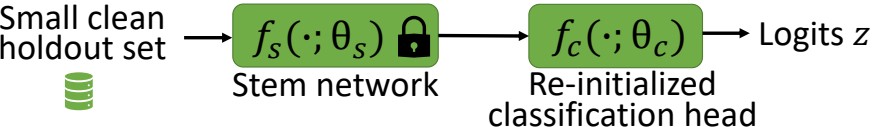

Figure 1: Overview of our Trap and Replace strategy. Each block represents a subnetwork. The lock icon indicates that the subnetwork is fixed, otherwise it is trainable by default. Green subnetworks are defended or trained only using clean samples (and thus are hardly infected by backdoor samples). The red one is the trap subnetwork used to bait and trap the backdoor attacks (and thus are infected).

backdoor triggers can mislead the network to learn an undesired strong correlation between the trigger and the target label, which is termed the *backdoor correlation*. As a result, if the attacker adds the trigger pattern to a test sample, it will be classified as the target label, regardless of its ground truth class. In this way, the model's behavior on test samples can be controlled by the attacker with the added backdoor trigger.

Previous works have shown that such backdoor correlations are easy to learn but hard to forget (or unlearn) by DNNs [3, 4, 5]. For example, Liu *et al.* [3] and Gu *et al.* [6] showed that simply fine-tuning a backdoored model on a small portion of clean samples is hardly effective to forget the already learned backdoor correlations. Li *et al.* [4] enhanced the above fine-tuning method with an extra attention distillation step. However, the performance of the new method in [4] is particularly sensitive to the type of underlying attack and data augmentation techniques [5]. Retraining a small portion of the network from scratch is also not effective (to be shown in our experiments), since the entire network may have been affected by the backdoor training samples. Retraining the entire or a large portion of the network from scratch using a small number of clean samples may succeed in removing the backdoor correlations, but it will significantly hurt the model performance since a huge amount of data is required to train DNNs from scratch.[1] Some previous works tried to identify and prune the neurons which are most heavily infected by backdoor training samples [3, 11]. However, the identification results for such "infected neurons" are noisy and can empirically fail as shown in [12, 5] (to be shown in our experiments, too). In summary, the challenge raises from the high-level freedom in model training: the backdoor samples can potentially infect any neurons in the entire network. With a limited amount of clean training samples, it is challenging for the defender to precisely locate and fix all those infected neurons.

To address this challenge, we propose a novel backdoor defense strategy named *Trap and Replace* (T&R), which makes it much easier to remove the learned backdoor correlations from the network. In a nutshell, T&R first baits and traps the backdoors in a small and easy-to-replace subnetwork, and then replaces the poisoned small subnetwork (i.e., the bait subnetwork) with an untainted one re-trained from scratch using a small amount of clean data.

As illustrated in Figure 1, this strategy has two stages. In the first stage (i.e., the bait-and-trap stage), we first divide the classification model into two subnetworks: a stem taking up most of the parameters and a light-weighted classification head. We then add an auxiliary head on top of the shared stem network to conduct an image reconstruction task. The entire model is trained end-to-end by jointly

---

[1] Data-efficient training such as semi-supervised learning and self-supervised learning are not naive solutions, since themselves are vulnerable to backdoor attacks [7, 8, 9, 10].

optimizing on the two tasks (i.e., image classification and reconstruction) using the poisoned training set. With the auxiliary image reconstruction task, backdoors can be effectively baited towards and trapped into the light-weighted classification head, while the shared stem is prevented from overfitting to the backdoor features. The **intuition** is that the auxiliary image reconstruction task encourages the stem network to keep sufficient low-level visual features that are hard-to-learn but semantically correct, protecting the stem network from overfitting to the easy-to-learn but semantically incorrect backdoor correlations. In contrast, we apply no defense mechanism on the light-weighted classification head, leaving it more vulnerable than the stem network. As a result, when trained on backdoored datasets, the backdoors are easily baited towards and trapped in the unprotected classification head, leaving the stem network hardly poisoned.

In the second stage, we replace the poisoned light-weighted classification head with an untainted one, as illustrated in Figure 1. Specifically, we re-initialize the classification head to random values, and then train it from scratch on a small holdout dataset with clean samples. It is feasible to re-train the classification head from scratch using only a small amount of clean samples, because it is light-weighted (e.g., only two convolutional and one fully connected layers in our experiments) and the shared stem obtained in stage 1 can already extract high-quality deep features. As a result, both the stem and the classification head in the final network are hardly affected by backdoor training samples.

We evaluate the effectiveness of Trap and Replace on three image classification datasets and against ten different backdoor attacks. Experimental results show Trap and Replace outperforms previous state-of-the-art methods by up to 20.57%, 9.80%, and 13.72% attack success rate (ASR) and in-average 3.14%, 1.80%, and 1.21% clean classification accuracy (CA) on CIFAR10, GTSRB, and ImageNet-12, respectively. We further show our method is robust to potential adaptive attacks, where the attacker is aware of the applied defense strategy and able to take countermoves.

## 2 Related work

### 2.1 Backdoor attack methods

Liu *et al.* [1] first successfully conducted backdoor/trojan attacks on DNNs, by adding pre-defined backdoor triggers (e.g., a square patch with a fixed pattern) onto the images and modifying the corresponding labels to the attacker-desired target label. Gu *et al.* [6] later showed that backdoor attacks can be successfully preserved during transfer learning. In other words, backdoors injected in pre-trained models can be transferred to downstream tasks. Yao *et al.* [13] proposed the latent backdoor attack (LBA), which targets the hidden layers instead of the output layer, so that the attack can better survive downstream transfer learning. Chen *et al.* [14] showed that backdoor images can also be generated by blending the backdoor trigger image with the clean images. The early backdoor attacks have two limitations making them potentially hard to survive careful human inspection. The first is that the triggers are usually small but still visible to humans. The second limitation is that traditional backdoor attacks are dirty-label backdoor attacks: they change the labels of backdoor training samples to the attacker-desired targeted label, leading to inconsistency between the content of the sample and its label.

To overcome the first limitation, Zhong *et al.* [15], Li *et al.* [16], and Li *et al.* [17] proposed invisible backdoor attacks which add small and invisible perturbations as backdoor triggers to clean images. Nguyen and Tran [18] used imperceptible warping-based triggers to bypass human inspection. More recently, Zeng *et al.* [19] proposed to generate smooth backdoor triggers using frequency information to prevent the severe high-frequency artifacts of previous attack methods.

To overcome the second limitation, clean-label backdoor attacks (i.e., attacks that do not require to modify the labels of backdoor training samples) have been proposed [20, 21, 22]. For example, Barni *et al.* [21] added ramp signals as the backdoor trigger to the images of the target class, without modifying the labels. The intuition of such attack is that the model tends to overfit the easy-to-learn ramp signals instead of the semantically meaningful but hard-to-learn object visual features. Thus, a strong backdoor correlation between the ramp signal and the target class will be learned by the model. With a similar underlying intuition, label-consistent backdoor attack (LCBA) [20] adds both adversarial noises and simple patch patterns onto the backdoor training images. This makes the semantically-correct visual features in backdoor training images hard to learn, since they are perturbed by adversarial noises. As a result, the model will focus on overfitting the easy-to-learn backdoor correlation (i.e., the strong correlation between the backdoor pattern and the ground truth

label) from the backdoor training samples. Zeng *et al.* [22] proposed a more practical clean-label backdoor attack which requires less prior information of the training set. Hidden trigger attacks [23] enjoy the benefits of both sides as a clean-label attack with invisible triggers.

## 2.2 Backdoor defense methods

Backdoor defense methods aim to obtain clean models without backdoors when trained on potentially poisoned data. As one of the earliest works on defending backdoor attacks, Tran *et al.* [24] proposed an outlier removal method based on the spectral signature to distinguish and remove backdoor samples from clean samples. Hayase *et al.* [25] improved upon [24] by using a more robust outlier removal method. Other outlier detection methods such as activation clustering [26], prediction consistency [27], and influence function [28] have also been used to detect backdoor samples.

In [3], a concurrent work with [24], the authors proposed another earliest backdoor defense method named Fine-Pruning (FP). Motivated by the empirical finding that clean and backdoor samples tend to activate different neurons, FP first prunes the neurons that are dormant on clean samples and then fine-tune the pruned network on a small number of clean samples. Later works in this line make improvements on locating the backdoor neurons (i.e., the neurons that are activated by backdoor samples but not clean samples) [11, 29]. Adversarial neuron perturbations (ANP) [11] prunes the sensitive neurons under adversarial perturbations to improve backdoor robustness. A very recent work [29] used Shapley value as a measure to prune backdoor neurons.

Robust training methods have also been used to prevent the learning of semantically-incorrect backdoor correlations during model training [30, 31, 32]. For example, Du *et al.* [30] used differentially private training [33] to prevent the learning of backdoor correlations. Using strong data augmentation methods such as MixUp [34] and MaxUp [35] can also benefit backdoor defense [31]. Self-supervised pre-training achieves promising results against backdoor attacks developed for supervised learning [32]. However, more recent works have successfully developed backdoor attacks tailored for self-supervised learning and contrastive learning [9, 10]. Adversarial training, which is originally proposed to improve model robustness against adversarial attacks [36, 37], has also been adapted to empirically improve robustness against backdoor attacks [38]. A recent work [39] further theoretically showed that backdoor filtering and adversarial robust generalization are nearly equivalent under assumptions.

Neural cleanse [40] set up the starting point for a new line of research [41, 42, 43]. This line of methods first inverts engineer the unknown backdoor trigger from the poisoned models, and then unlearns the backdoor using the synthesized trigger. A recent work [5] made improvements over the above methods by using a novel unlearning method that requires fewer presumptions about the backdoor trigger. As a result, the proposed method, named implicit backdoor adversarial unlearning (I-BAU), successfully defends a wide range of backdoor attacks with different trigger patterns. In contrast, previous methods in [41, 42, 43] all have failure cases, as shown in [5].

Recently, Li *et al.* [12] proposed a novel backdoor dense method named anti-backdoor learning (ABL), which largely outperforms previous methods. Specifically, ABL uses a novel local gradient ascent loss (LGA) to isolate backdoor examples from clean training samples: Using the LGA loss, backdoor training samples will have statistically lower loss values than clean training samples. As a result, a small amount of backdoor samples can be successfully isolated, which are further used to unlearn the backdoor correlations. One limitation of ABL, as discussed in the original paper, is that the loss value can be a noisy measure to distinguish backdoor samples under certain cases. Also ABL requires careful hyper-parameter tuning [12]. There is also another line of works focusing on detecting backdoor-infected models [44, 45, 46, 47, 48]. Their main goal is to predict whether a given model is infected by backdoor attacks, instead of preventing the learning of backdoor correlations.

Our T&R is a brand-new backdoor defense strategy that does not fall into any of the above categories. Among all categories, T&R is most relevant with the pruning-based methods [3, 11, 29]: we share the same ultimate goal to remove infected neurons. However, unlike those methods which spend much effect in locating the infected neurons, our method take the initiative to set a trap in the model to bait and trap the backdoor. As a result, we do not need to locate the infected neurons, since we know exactly where the trap is set. The only thing we need is to replace the infected trap (i.e., a light-weighted subnetwork) with an untainted one trained on a small clean dataset.

More related works are discussed in Appendix A.

# 3 Method

In this section, we first describe the problem setting and define the notations in Section 3.1. We then formally present our proposed method in Section 3.2.

## 3.1 Problem setting and notations

We consider the application scenario where the defender collects training data from untrusted sources (e.g., uploaded by untrusted users or from the Internet) which potentially contains backdoor samples, and then trains the model using the collected data. The goal of the defender is to obtain a clean model using the collected backdoored dataset, with the help of a small clean holdout set from the same distribution of the training set.[2]

Specifically, the training set $\mathcal{D}_{\text{train}}$ contains an unknown portion of backdoor training samples. We define the poison ratio $\alpha$ as the percentage of the poisoned training samples in the entire training set $\mathcal{D}_{\text{train}}$. Following previous works [3, 4, 11, 5], we assume that the defender has access to a small holdout set $\mathcal{D}_{\text{h}}$ with clean samples (i.e., samples without backdoor triggers). For evaluation, we have two test sets: a clean test set $\mathcal{D}_{\text{test}}^{\text{clean}}$ with only clean test samples, and a backdoor test set $\mathcal{D}_{\text{test}}^{\text{bd}}$ with backdoor test samples generated by applying backdoor patterns onto the samples in $\mathcal{D}_{\text{test}}^{\text{clean}}$. The goal of backdoor defense methods is to reduce the attack success rate (ASR) on $\mathcal{D}_{\text{test}}^{\text{bd}}$, while keeping high clean accuracy (CA) on $\mathcal{D}_{\text{test}}^{\text{clean}}$.

As illustrated in Figure 1, our framework has three subnetworks: a stem network $f_s$ with parameters $\theta_s$, a light weighted classification head $f_c$ with parameters $\theta_c$, and an auxiliary image reconstruction head $f_r$ with parameters $\theta_r$. For an input image $x$, the stem network outputs a hidden feature $h(x) = f_s(x; \theta_s)$. The classification head outputs the classification logits $z(x) = f_c(h(x); \theta_c)$. The image reconstruction head outputs the reconstructed image $\hat{x}(x) = f_r(h(x); \theta_r)$.

## 3.2 The proposed method

Our Trap and Replace (T&R) backdoor defense strategy consists of two stages. Below we formally describe the process of each stage.

**Stage 1: Bait and trap the backdoor into the classification head.** As intuitively explained in Section 1, we use the auxiliary image reconstruction task to regularize the stem network $f_s$ to keep enough semantically correct low-level visual features. The stem network is thus protected from overfitting to the easy-to-learn but semantically incorrect backdoor correlations.[3] In contrast, we apply no defense mechanisms on the light-weighted classification head $f_c$, leaving it more vulnerable than the stem. As a result, the backdoor attacks are easily baited towards and trapped in the classification head, leaving the stem network hardly infected.

Specifically, we jointly train the classification task and the auxiliary image classification task on the poisoned training set $\mathcal{D}_{\text{train}}$. The entire model is updated end-to-end. More formally, we solve the following optimization problem:

$$\min_{\theta_s, \theta_c, \theta_r} \mathbb{E}_{x \sim \mathcal{D}_{\text{train}}} \mathcal{L}_{\text{clf}}(x) + \lambda_1 \mathcal{L}_{\text{rec}}(x), \tag{1}$$

where $\mathcal{L}_{\text{clf}}(x)$ and $\mathcal{L}_{\text{rec}}(x)$ are the image classification and reconstruction losses, respectively, and $\lambda_1$ is the trade-off weight between the two loss terms. The image classification loss $\mathcal{L}_{\text{clf}}(x)$ is simply the cross-entropy loss between logits $z(x)$ and corresponding ground truth label $y(x)$. The reconstruction loss is the $\ell_2$ loss between the original and reconstructed images with the total variation regularization: $\mathcal{L}_{\text{rec}}(x) = \|\hat{x}(x) - x\|_2 + \lambda_2 \text{TV}(\hat{x}(x))$, where $\text{TV}(\cdot)$ is the total variation function, which is widely used to improve the spatial smoothness and visual quality of the reconstructed image $\hat{x}(x)$ [49, 50], and $\lambda_2$ is the loss trade-off weight.

---

[2]The relation and difference in application scenario with previous works are discussed in Appendix D.

[3]In contrast, the stem network obtained from normal training (i.e., without the auxiliary image reconstruction task) will overfit to the semantically incorrect but easy-to-learn backdoor correlations. See our ablation study in Section 4.4 (Table 4) for details.

**Stage 2: Replace the infected classification head.** After stage 1, the stem network keeps enough semantically correct low-level visual features. Now the missing piece of a disinfected image classifier is an untainted light-weighted classification head, which can be obtained by re-training $f_c$ from scratch using the small clean holdout set $\mathcal{D}_h$. Specifically, we first discard the auxiliary image reconstruction head $f_r$, and re-initialize the classification head parameters $\theta_c$ to random values. We then re-train $\theta_c$ on $\mathcal{D}_h$, while keeping the stem parameters $\theta_s$ fixed to the values learned in stage 1. More formally, we solve the below optimization problem:

$$\min_{\theta_c} \mathbb{E}_{x \sim \mathcal{D}_h} \mathcal{L}_{\text{clf}}(x). \tag{2}$$

Note that solving Eq. (2) from scratch is feasible since $f_c$ is light-weighted. To further prevent overfitting on the small $\mathcal{D}_h$, we apply Dropout (with $50\%$ ratio) [51] and label smoothing (with smoothing factor 0.1) [52] as regularization when solving Eq. (2). Finally, we summarize the workflow of T&R in Algorithm 1.

---

**Algorithm 1:** Trap and Replace (T&R)

---

**Input:** Poisoned training set $\mathcal{D}_{\text{train}}$, a small holdout set $\mathcal{D}_h$.
**Output:** A disinfected image classifier $f_c(f_h(\cdot; \theta_h); \theta_c)$.

1  // Stage 1:
2  Randomly initialize $\theta_h$, $\theta_c$, and $\theta_r$.
3  Optimize $\theta_h$, $\theta_c$, and $\theta_r$ on $\mathcal{D}_{\text{train}}$ according to Eq. (1).
4  // Stage 2:
5  Randomly initialize $\theta_c$.
6  Optimize $\theta_c$ on $\mathcal{D}_h$ according to Eq. (2).

---

## 4 Experiments

### 4.1 Experimental settings

**Datasets and models** Following [12, 5, 32], we conduct experiments on CIFAR10 [53], GTSRB [54], and ImageNet-12 (a subset of ImageNet [55] with 12 classes); we use WideResNet (WRN16-1) [56] on the first two datasets and ResNet34 [57] on ImageNet-12.

**Backdoor attacks** We evaluate the effectiveness of our method against 10 different backdoor attacks. Specifically, we use all 7 attacks in [5]: BadNet with white square pattern (denoted as BadNet-White) [6], Blend [14], $\ell_0$-Invisible [16], $\ell_2$-Invisible [16], Smooth [19], Trojan with square pattern (denoted as Trojan-SQ) [1], and Tojan with watermark pattern (denoted as Trojan-WM) [1]. We further include 3 more attacks used in [12]: BadNet with grid square pattern (denoted as BadNet-Grid) [6], label-consistent backdoor attack (LCBA) [20], and SIG [21]. Note that LCBA and SIG are clean-label attacks and all others are dirty-label attacks. Following [12], we set poison ratio $\alpha = 10\%$ for all attacks (i.e., 5000 training images are poisoned in CIFAR10). Detailed settings and visualization for each attack are shown in Appendix B.1. As pointed out by [12], some attacks fail to be reproduced following their original papers on GTSRB and Imagenet-12. Following the suggestions in [12], we omit CLBA attack on GTSRB and use four attacks on ImageNet-12.

**Baseline methods** We compare our method with six baseline methods: normal training (denoted as "No defense"), differentially private training (DP) [30], NAD [4], ANP [11], ABL [12], and I-BAU [5]. The last three are recent state-of-the-art methods that have been shown to largely outperform previous baselines. The results of DP are only shown in Appendix due to space limit (and also since it achieves the least competitive results). We omit ANP and I-BAU on ImageNet-12, since the original papers only provided experimental settings on CIFAR10 and we fail to produce reasonable results on ImageNet after careful hyper-parameter tuning.

**Evaluation measures** Following [12, 5], we use attack success rate (ASR) and clean accuracy (CA) as the evaluation measures. ASR is defined as the percentage of backdoor test samples that can successfully fool the model to the target class desired by the attacker. CA is simply the classification accuracy on a clean test set. The smaller ASR ($\downarrow$) and the larger CA ($\uparrow$) indicate better backdoor defense performance.

**Defense settings** Previous works usually use $10\%$ of the training set as the clean holdout set [3, 4]. In this paper, we consider a more challenging setting where less clean holdout data are available for defenders. Specifically, for all methods requiring a clean holdout set (i.e., NAD, ANP, I-BAU, and our T&R), we use $5\%$ of the training set as the clean holdout set on CIFAR10 and GTSRB. The ratio is still set to $10\%$ on ImageNet-12 for all methods since it is a more challenging dataset. Please see Appendix B for more details on experimental settings.

## 4.2 Main results

Table 1: Results on CIFAR10 using WRN16-1. The best results are shown in bold. All numbers are reported in percentages.

| Attack method | No defense | | NAD | | ANP | | ABL | | I-BAU | | Ours | |
|---|---|---|---|---|---|---|---|---|---|---|---|---|
| | ASR | CA | ASR | CA | ASR | CA | ASR | CA | ASR | CA | ASR | CA |
| BadNet-Grid | 99.98 | **88.36** | 8.60 | 80.23 | 2.64 | 84.55 | 3.20 | 86.35 | 8.24 | 81.34 | **1.21** | 84.42 |
| BadNet-White | 96.91 | **88.87** | 12.80 | 79.68 | 3.68 | 85.74 | 9.32 | 80.32 | 8.80 | 86.17 | **3.14** | 83.96 |
| Blend | 99.11 | **88.95** | 10.56 | 81.89 | **5.62** | 82.35 | 19.91 | 80.63 | 12.27 | 75.18 | 10.59 | 83.82 |
| $\ell_0$-Invisible | 99.93 | **89.36** | 30.25 | 76.84 | 15.76 | 72.5 | 16.76 | 76.73 | 9.21 | 81.15 | **2.91** | 84.04 |
| $\ell_2$-Invisible | 100.00 | **89.28** | 10.06 | 75.63 | 22.97 | 70.52 | 7.49 | 76.82 | 2.64 | 82.48 | **0.74** | 84.01 |
| Smooth | 98.14 | **89.24** | 15.68 | 79.10 | 33.42 | 77.49 | 20.21 | 70.79 | 24.80 | 67.93 | **4.23** | 83.63 |
| Trojan-SQ | 98.23 | **89.64** | 12.52 | 80.56 | 23.84 | 72.06 | **3.16** | 80.44 | 16.61 | 83.27 | 6.54 | 79.92 |
| Trojan-WM | 99.61 | **89.48** | 33.64 | 79.60 | 18.04 | 75.66 | 7.31 | 84.89 | **4.55** | 85.30 | 12.66 | 79.97 |
| SIG | 99.93 | **89.27** | 3.20 | 75.24 | 0.05 | 85.67 | 8.79 | 60.07 | 1.89 | 77.05 | **0.02** | 82.97 |
| LCBA | 90.42 | **86.35** | 10.64 | 77.82 | **1.02** | 84.21 | 6.54 | 78.56 | 8.88 | 78.05 | 5.41 | 82.57 |
| Average | 97.95 | **88.54** | 14.79 | 78.66 | 12.70 | 79.08 | 10.27 | 77.56 | 9.79 | 79.79 | **4.75** | 82.93 |

Table 2: Results on GTSRB. The best results are shown in bold. All numbers are reported in percentages.

| Attack method | No defense | | NAD | | ANP | | ABL | | I-BAU | | Ours | |
|---|---|---|---|---|---|---|---|---|---|---|---|---|
| | ASR | CA | ASR | CA | ASR | CA | ASR | CA | ASR | CA | ASR | CA |
| BadNet-Grid | 100 | **96.36** | 1.62 | 93.56 | 8.97 | 95.44 | 4.11 | 95.52 | 5.56 | 95.36 | **0.20** | 95.94 |
| BadNet-White | 96.32 | **96.11** | 0.96 | 90.61 | 12.68 | 92.06 | **0.00** | 95.14 | 6.63 | 95.55 | 0.01 | 95.74 |
| Blend | 99.95 | **96.50** | 5.64 | 91.26 | 20.36 | 90.53 | 4.94 | 92.61 | **0.00** | 82.03 | 1.98 | 95.62 |
| $\ell_0$-Invisible | 100 | **95.50** | 25.60 | 91.35 | 34.96 | 88.54 | **0.60** | 96.39 | 4.73 | 94.31 | 1.32 | 95.87 |
| $\ell_2$-Invisible | 99.93 | **96.81** | 12.43 | 72.69 | 24.83 | 90.41 | 100 | 94.74 | 9.83 | 94.60 | **0.03** | 96.10 |
| Smooth | 100 | **96.58** | 33.20 | 75.74 | 37.95 | 82.56 | 32.24 | 76.70 | 2.68 | 94.74 | **0.11** | 96.12 |
| Trojan SQ | 97.75 | **96.53** | 1.52 | 90.56 | 12.06 | 90.74 | **0.45** | 95.88 | 5.71 | 94.49 | 1.47 | 95.17 |
| Trojan WM | 100 | **96.68** | 1.58 | 92.26 | 25.68 | 88.64 | **0.55** | 95.41 | 7.47 | 95.30 | 7.06 | 91.95 |
| SIG | 87.51 | **96.32** | 50.68 | 88.56 | 20.60 | 82.34 | 66.80 | 96.50 | 10.04 | 94.53 | **0.54** | 94.56 |
| Average | 97.94 | **96.38** | 16.45 | 87.40 | 22.01 | 89.03 | 23.30 | 93.21 | 5.85 | 93.43 | **1.41** | 95.23 |

Table 3: Results on ImageNet-12. The best results are shown in bold. All numbers are reported in percentages.

| Attack method | No defense | | NAD | | ABL | | Ours | |
|---|---|---|---|---|---|---|---|---|
| | ASR | CA | ASR | CA | ASR | CA | ASR | CA |
| BadNet-Grid | 99.67 | **74.17** | 13.67 | 71.67 | 3.59 | 74.06 | **0.67** | 72.64 |
| Blend | 99.17 | **74.83** | 33.50 | 70.17 | 25.06 | 70.12 | **11.33** | 67.33 |
| Trojan-WM | 100 | **77.33** | 29.33 | 73.33 | **3.26** | 72.08 | 7.52 | 70.83 |
| SIG | 86.00 | 71.67 | 19.67 | 70.83 | 7.67 | 67.01 | **5.33** | **77.33** |
| Average | 96.21 | **74.50** | 24.04 | 71.50 | 9.89 | 70.82 | **6.21** | 72.03 |

The main results on CIFAR10, GTSRB, and ImageNet-12 are shown in Table 1, 2, and 3, respectively. As we can see, our method largely outperforms previous state-of-the-art methods in terms of both

ASR and CA on all three datasets. Specifically, on CIFAR10, our method outperforms I-BAU (the most competitive baseline method) by 20.57% ASR against Smooth attack, 10.07% ASR against Trojan-SQ attack, and 3.14% average CA. On GTSRB dataset, our method outperforms I-BAU by 9.80% ASR against $\ell_2$-Invisible attack, 9.50% ASR against SIG attack, and 1.80% average CA. On ImageNet-12 dataset, our method outperforms ABL by 13.72% ASR against Blend attack, 3.68% average ASR, and 1.21% average CA.

Moreover, we can observe that our method has almost no failure cases on all three datasets. In contrast, all baseline methods have failure cases, with either a significant drop in CA compared with normal training (i.e., "No defense") or an unacceptably high ASR. For example, ABL fails on $\ell_2$-Invisible attack and SIG attack on GTSRB, and I-BAU fails on Smooth attack on CIFAR10 and Blend attack on GTSRB.

### 4.3 Is the stem network really backdoor-free?

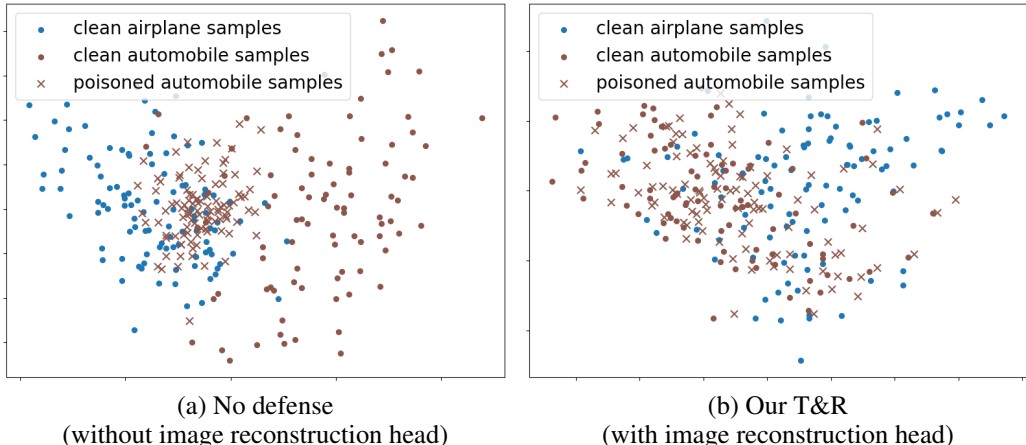

(a) No defense
(without image reconstruction head)

(b) Our T&R
(with image reconstruction head)

Figure 2: Output feature scatters of the stem network $f_s$ when trained without (left) and with (right) the auxiliary image reconstruction head. Principle component analyses (PCA) is applied to project the features to two-dimensional space. Three types of test samples are visualized: clean "airplane" samples, clean "automobile" samples, and backdoored "automobile" samples (i.e., automobile images with $\ell_2$-Invisible pattern). Both models are trained on CIFAR10 dataset poisoned by $\ell_2$-Invisible attack which aims to map the backdoor pattern to the target "airplane".

In this section, we verify our assumption that the image reconstruction task can protect the stem network from overfitting to the backdoor correlations, by visualizing the feature scatters. Specifically, we compare the output features of the stem network $f_s$ when trained with or without the auxiliary image reconstruction task. We visualize the feature distributions of clean "airplane" samples, clean "automobile" samples, and poisoned "automobile" samples via PCA in Figure 2.

As shown in Figure 2(a), without the auxiliary image reconstruction task, the stem network $f_s$ maps poisoned "automobile" samples to the feature space of clean "airplane" samples instead of the clean "automobile" feature space. This demonstrates that $f_s$ has learned the backdoor correlation: It ignores the semantic features related to "automobile" in the poisoned "automobile" samples, but overfits the semantically-incorrect backdoor correlations between the backdoor pattern and the backdoor target "airplane". In contrast, as shown in Figure 2(b), when trained with the image reconstruction head, clean and poisoned "automobile" samples are projected to similar feature space, which is different from the "airplane" feature space. This indicates that $f_s$ learns semantically-correct features, instead of the backdoor correlations, on the poisoned samples. In conclusion, the visualization results show that the auxiliary image reconstruction task successfully hinders the stem network from learning the backdoor correlations.

## 4.4 Ablation study

**The importance of the auxiliary image reconstruction task**   In this paragraph, we show the importance of two building blocks in T&R: the auxiliary image reconstruction task in stage 1, and the re-training of the classification head in stage 2. The results are shown in Table 4. As shown in the first row, if we remove the auxiliary image reconstruction task in stage 1, the model will not effectively unlearn the backdoor correlations even we retrain the classification head from scratch. For example, the ASRs of $\ell_2$-Invisible and Trojan-WM are still as high as $99.99\%$ and $51.87\%$, respectively. This indicates that without the auxiliary image reconstruction task, the backdoor training samples have infected a large portion of the network, including both the classification head and the stem network. In contrast, with the help of the auxiliary image reconstruction task, we successfully trapped the backdoor correlations into the light-weighted classification, while leaving the stem network largely uninfected. On the other hand, as shown in the second row in Table 4, using the auxiliary image reconstruction task alone will not directly reduce ASR, since the classification head is still infected. In conclusion, both building blocks are necessary: The defense will fail with either one missing.

Table 4: Ablation study on the building components of T&R using CIFAR10. In the last row, we report the mean and standard deviation of the results over three random runs of our method.

| Auxiliary image reconstruction task | Replace classification head | $\ell_2$-Invisible ASR | CA | Trojan-WM ASR | CA | SIG ASR | CA | LCBA ASR | CA |
|:---:|:---:|:---:|:---:|:---:|:---:|:---:|:---:|:---:|:---:|
| ✗ | ✓ | 99.99 | 85.08 | 51.87 | 84.26 | 26.96 | 84.03 | 45.81 | 84.31 |
| ✓ | ✗ | 99.96 | 89.06 | 99.81 | 88.50 | 97.24 | 81.10 | 92.46 | 81.22 |
| ✓ | ✓ | **0.72** ±0.09 | 83.72 ±0.25 | **12.01** ±1.16 | 80.01 ±0.33 | **0.01** ±0.01 | 82.99 ±0.03 | **5.37** ±0.45 | 82.47 ±0.11 |

We also report the error bars of our method in the last row of Table 4. Specifically, we run our method three times with different random seeds and report the mean ($\mu$) and standard deviation ($\sigma$) of ASR and CA in the form of $\mu \pm \sigma$. As we can see, the performance of our method is statistically stable under all four different attacks.

**From which layer should we separate the stem and classification head?**   The ablation study results on the stem-head separation layer are shown in Table 5. As we can see, if the stem contains too many layers (e.g., as in the first row), the backdoor defense will fail, since it is hard to trap all the backdoors in too tiny a classification head (e.g., only one fully connected layer as in the first row). On the other hand, if the head contains too many layers (e.g, as in the third row), the backdoor defense will be successful, but CA will significantly drop. This is because, with a limited amount of holdout clean samples, it is infeasible to re-train too large a classification head from scratch. In summary, a properly sized classification head is important for the practical success of our method.

Table 5: Ablation study on the stem-head separation layer. Experiments are conducted on CIFAR10 with WRN16-1 backbone, which has a total of 13 convolutional and 1 fully connected (FC) layers. In the first row, $f_c$ has only one FC layer, and all convolutional layers belong to $f_s$. The second row is our default setting, where $f_c$ has the last two convolutional layers and the FC layer. The best results are shown in bold and the second-best ones are underlined.

| Number of layers $f_s$ | $f_c$ | Blend ASR | CA | $\ell_2$-Invisible ASR | CA | Smooth ASR | CA | Trojan-WM ASR | CA |
|:---:|:---:|:---:|:---:|:---:|:---:|:---:|:---:|:---:|:---:|
| 13 | 1 | 97.41 | **88.04** | 99.99 | **87.93** | 96.02 | **87.60** | 99.63 | **87.52** |
| 11 | 3 | 10.59 | 83.82 | 0.74 | 84.01 | 4.23 | 83.63 | 12.66 | 79.97 |
| 9 | 5 | **3.01** | 74.99 | **0.37** | 73.50 | **4.17** | 73.96 | 13.83 | 77.36 |

**Ablation study on poison ratio**   In this paragraph, we study the performance of T&R under different poison ratios. Specifically, we set the poison ratio $\alpha$ to $5\%$, $10\%$ (the default value used in our main experiments), and $20\%$, which are the popular values used in previous works [12, 5]. We also compare T&R with I-BAU, which is the most competitive baseline as shown in Table 1, under these different poison ratios. The results are summarized in Table 6. As we can see, the advantage of our method holds across different poison ratios.

Table 6: Ablation study on the poison ratio $\alpha$ using CIFAR10. The best results are shown in bold.

| $\alpha$ | Defense method | Blend | | $\ell_2$-Invisible | | Smooth | | Trojan-WM | |
| | | ASR | CA | ASR | CA | ASR | CA | ASR | CA |
| --- | --- | --- | --- | --- | --- | --- | --- | --- | --- |
| 5% | Ours | **9.57** | **83.52** | **0.54** | **84.21** | **2.04** | **83.96** | 7.61 | 80.12 |
| | I-BAU | 10.16 | 81.96 | 1.68 | 81.04 | 8.58 | 80.92 | **5.34** | **82.69** |
| 10% | Ours | **10.59** | **83.82** | **0.74** | **84.01** | **4.23** | **83.63** | 12.66 | 79.97 |
| | I-BAU | 12.27 | 75.18 | 2.64 | 82.48 | 24.80 | 67.93 | **4.55** | **85.30** |
| 20% | Ours | **8.74** | **82.15** | **0.84** | **84.25** | **3.51** | **82.96** | 14.28 | 79.84 |
| | I-BAU | 15.57 | 81.06 | 4.38 | 83.06 | 16.69 | 80.44 | **6.94** | **83.33** |

More experimental results, including results against potential adaptive attack, ablation study on hyper-parameters, ablation study on the size of clean holdout set $\mathcal{D}_h$, and results on non-poisoned datasets (i.e., when poison ratio $\alpha = 0$), are presented in Appendix C.

## 5 Discussion

The **major finding** in this paper is that the auxiliary image reconstruction loss can successfully trap backdoors into a small subnetwork while preserving the rest of the network largely uncontaminated, as illustrated in Figure 2. The **impact** of this finding is that it provides a promising ***decrease-and-conquer*** strategy for backdoor defense. Since the stem network is largely uncontaminated, defenders only need to tackle the small backdoored subnetwork, which is easier than sanitizing the entire network. This finding inspires multiple **future work** directions. First, our "trapping" strategy can potentially be combined with previous pruning-based backdoor neuron removal methods [11, 29] for better results. This is because it reduces the search space of potential backdoor neurons: We only need to search and remove backdoored neurons in the small backdoored subnetwork, instead of the entire network as done previously in [11, 29]. Second, although sufficiently effective against existing backdoor attacks, the image reconstruction loss is not guaranteed to be the best option for the "trapping" strategy, and one may research better alternatives as future work.

**Limitation**   Our method has less flexible application scenario compared with previous backdoor defense methods, since it requires the backdoored training set. Most previous defense methods [3, 4, 5] take the backdoored model, instead of the original backdoored training set, as input. Unlike those methods, our method cannot sanitize the backdoored models when the backdoored training set is not available. With that said, the application scenario of our method is still popular in real-world applications. Please see Appendix D for details.

## 6 Conclusion

In this paper, we show that an auxiliary image reconstruction loss can successfully hinder the learning of backdoor attacks in image classification tasks. Based on this observation, we propose a brand-new backdoor defense strategy named Trap and Replace (T&R). Empirical results on three image classification datasets show the advantage of our method over previous state-of-the-art methods.

## Acknowledgement

This material is based in part upon work supported by the National Science Foundation under Grant IIS-2212174, IIS-2212176, IIS-1749940, and Office of Naval Research N00014-20-1-2382.

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
