# A    More related works

In this section, we discuss more related works in addition to those in Section 2.

Recently, self-supervised learning has also been shown to be vulnerable to backdoor attacks [9, 10]. Yan *et al.* [7] and Carlini [8] successfully designed backdoor attacks against semi-supervised learning. It has been shown that backdoored models can be obtained in the near vicinity of clean models, making it harder to detect backdoored models from clean models [58, 59]. Another line of research studies on deployment-stage backdoor attacks [60, 61, 62], which inject backdoors into pre-trained models by perturbing the weights, instead of training them on backdoor samples as in traditional training-stage backdoor attacks [1, 6]. In this work, we focus on defending training-stage backdoor attacks.

# B    More implementation details

In this section, we provide more details on our experimental settings, in addition to those in Section 4.1.

## B.1    Backdoor attack details

For BadNet-Grid, Trojan-SQ, and SIG, we use the triggers provided in the official codes of [12][4]. For LCBA attack, we use the official poisoned dataset provided in the codes of the original paper[5]. For other attacks, we use the backdoor triggers provided in the official codes of [5][6]. The visualization of all 10 attacks are shown in Figure 3. Following [12], we set the target class to 0 on CIFAR10 and ImageNet-12, and 1 on GTSRB; we use all-to-one attack mode for all dirty-label attacks, where a portion of training samples from non-target classes are poisoned towards the target class; the poisoning ratio $\alpha$ is set to $10\%$ by default.

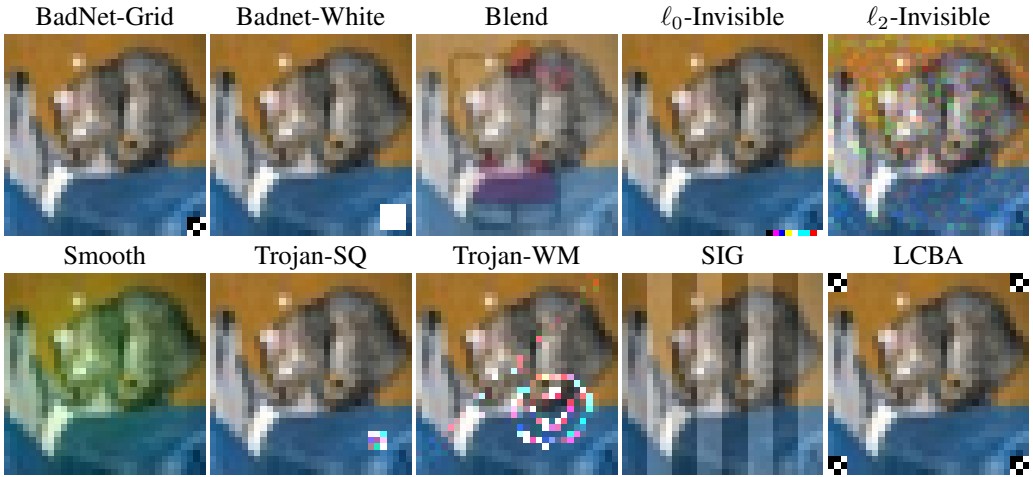

Figure 3: Visualization of different attacks on CIFAR10. The poisoned images are shown below the name of each attack.

## B.2    Backdoor defense details

For all defense methods, we use the same model structures and backdoor attack settings as described in Section 4.1 and Appendix B.1. Below we describe other detailed settings of each defense method.

**Normal training (i.e., "No defense")**    On CIFAR10 and GTSRB, we train for 200 epochs using Adam optimizer with initial learning rate $1 \times 10^{-3}$, cosine annealing learning rate scheduler, weight

---

[4]https://github.com/bboylyg/ABL

[5]https://github.com/MadryLab/label-consistent-backdoor-code

[6]https://github.com/YiZeng623/I-BAU

decay $5 \times 10^{-4}$, and batch size 256. On ImageNet-12, we train for 90 epochs using SGD optimizer with initial learning rate 0.1, cosine annealing learning rate scheduler, weight decay $5 \times 10^{-4}$, and batch size 256.

**Our method**  In stage 1, we set the loss trade-off parameters $\lambda_1 = 10$ and $\lambda_2 = 0.1$. Other hyper-parameters in stage 1 (e.g., learning rate, batch size, etc.) are set identical to those used in normal training. In stage 2, we use the same hyper-parameters as in stage 1, except that we set the batch size to 32 on CIFAR10 and GTSRB due to the small size of holdout set $\mathcal{D}_{\mathrm{h}}$.

**I-BAU**  The original I-BAU paper conducted experiments on a relatively small convolutional network. We empirically find the default hyper-parameters in their original paper do not lead to satisfying performance on WRN16-1, which is a more widely used model structure in backdoor defense papers [4, 12]. We turn the inner and outer learning rates of I-BAU, which are the two most important hyper-parameters, in $\{0.1, 1, 2, 5, 10\}$ and $\{1 \times 10^{-5}, 1 \times 10^{-4}, 1 \times 10^{-3}\}$, respectively. On both datasets, the best overall performance is achieved at 1 inner learning rate and $1 \times 10^{-4}$ outer learning rate, which we use to report the results of I-BAU.

**ANP**  Following the suggestion in the original paper [11], we tune the trade-off hyper-parameter between the natural and robustness loss in ANP on the discrete set $\{0.1, 0.2, 0.4, 0.5, 0.6\}$. The best overall performance is achieved at 0.2 on CIFAR10 and 0.1 on GTSRB.

**DP**  The original paper using DP for backdoor defense [30] conducted experiments on the simple MNIST dataset with a tiny three-layer convolutional neural network. Following [30, 5], we tune the noise multiplier in range $[0.5, 10]$ to achieve overall effectiveness across different attacks, and we keep the gradient clipping threshold to 1. In consistency with the results reported in [5], even with careful hyper-parameter tuning, DP fails to achieve satisfying results on datasets as complex as CIFAR10 and GTSRB.

**NAD and ABL**  Since our paper uses the same model structures with these two methods, we directly use the best hyper-parameters reported in their original papers for fair comparison.

### B.3  Hardware resources

All experiments are conducted on one NVIDIA RTX A6000 GPU.

## C  More experimental results

In this section, we provide more experimental results in addition to those in Section 4.

### C.1  Potential adaptive attack

In this section, we investigate the performance of our method under adaptive backdoor attacks that are intentionally designed to by-pass T&R. This is a more challenging setting for defenders, where the attacker is aware of the applied defense strategy and able to take countermoves. Since the core mechanism of T&R is to trap backdoor within the classification head and keep the stem network relatively clean, a potential adaptive attack is to intentionally inject backdoors into the stem network (i.e., the shallow or middle layers in the entire network).

Luckily, a previous work [13] has already designed an attack, named Latent Backdoor Attack (LBA), which serves the exact purpose. LBA is originally proposed to encode backdoor into hidden layers instead of output layers, so that the backdoor can survive transfer learning. It can also serve as the adaptive attack to our method. We try different settings denoted as LBA-$n$: LBA backdoor is injected to all layers before the $n$-th layer. For example, LBA-14 injects backdoor to the input of the last fully connected layer. As shown in Table 7, when equipped with the auxiliary image reconstruction task, our method can successfully defend LBA.

Table 7: Results of our method against the adaptive attack LBA-$n$ on CIFAR10.

| Auxiliary image reconstruction task | Replace classification head | LBA-14 | | LBA-12 | | LBA-10 | |
|:---:|:---:|:---:|:---:|:---:|:---:|:---:|:---:|
| | | ASR | CA | ASR | CA | ASR | CA |
| ✗ | ✓ | 93.64 | 82.06 | 90.62 | 84.12 | 84.65 | 83.49 |
| ✓ | ✓ | 4.32 | 84.52 | 4.56 | 84.67 | 0.68 | 85.43 |

## C.2 Ablation study on hyper-parameters

In this section, we show the performance of our method under different values of the hyper-parameters $\lambda_1$ and $\lambda_2$ in Eq. (1). Specifically, we first fix $\lambda_2 = 0.1$ and change $\lambda_1$ values, and then fix $\lambda_1 = 10$ and change $\lambda_2$ values. The results are shown in Table 8. As we can see, the ASR drops as the value of $\lambda_1$ increases from 0 to 10. This shows the effectiveness of our auxiliary image reconstruction task in defending against backdoor attack. As $\lambda_1$ goes beyond 10, the performance gain on ASR saturates, while the clean accuracy (CA) decreases. This is because too strong auxiliary loss on image reconstruction can bias the stem network towards learning image reconstruction features while ignoring the classification features. On the other hand, using a small but non-zero $\lambda_2$ also leads to better ASR than setting $\lambda_2 = 0$. This indicates that the total variation regularization can potentially prevent the stem network from learning some high-frequency features which commonly exist in backdoor samples [19], and thus helps defend backdoor attacks.

Alongside ASR and CA, we also show the mean square error (MSE) of the image reconstruction. Smaller MSE roughly indicates better image reconstruction quality. We also visualize image reconstruction results in Figure 4. As we can see, larger $\lambda_1$ values lead to better image reconstruction results.

Table 8: Ablation study on the hyper-parameters $\lambda_1$ and $\lambda_2$. Reported are results on CIFAR10 with $\ell_2$-Invisible attack.

| | $\lambda_1$ | | | | $\lambda_2$ | | |
|:---:|:---:|:---:|:---:|:---:|:---:|:---:|:---:|
| | 0 | 1 | 10 | 20 | 0 | 0.1 | 1 |
| ASR | 99.99 | 53.77 | 0.74 | 1.04 | 3.32 | 0.74 | 0.28 |
| CA | 85.08 | 83.70 | 84.01 | 81.37 | 82.46 | 84.01 | 83.51 |
| MSE | - | 0.0145 | 0.0085 | 0.0071 | 0.0097 | 0.0085 | 0.0091 |

## C.3 Ablation study on the size of clean holdout set

In this section, we show the performance of our method under different number of available clean training data (i.e., the size of the clean holdout set $\mathcal{D}_h$) in stage 2. Specifically, we use two different settings with $|\mathcal{D}_h| = 1250$ and $|\mathcal{D}_h| = 2500$ (i.e., 2.5% and 5% of the entire CIFAR10 training set, respectively). As shown in Table 9, our method can still success with even 2.5% clean training images available.

Table 9: Ablation study on the size of clean holdout set $\mathcal{D}_h$. Reported are results on CIFAR10 with $\ell_2$-Invisible attack. $|\mathcal{D}_h| = 2500$ (i.e., 5% of the entire training set) is the default setting used in our main experiments.

| $|\mathcal{D}_h|$ | 250 | 500 | 1250 | 2500 |
|:---:|:---:|:---:|:---:|:---:|
| ASR | 0.77 | 0.56 | 0.41 | 0.74 |
| CA | 72.00 | 76.99 | 83.31 | 84.01 |

## C.4 Results on non-poisoned datasets

The main purpose of the backdoor defense methods is to achieve good performance on poisoned training sets. However, in some practical situations, the model trainer might not know whether the training set is poisoned or not. A good solution for these situations is to first use backdoor detection

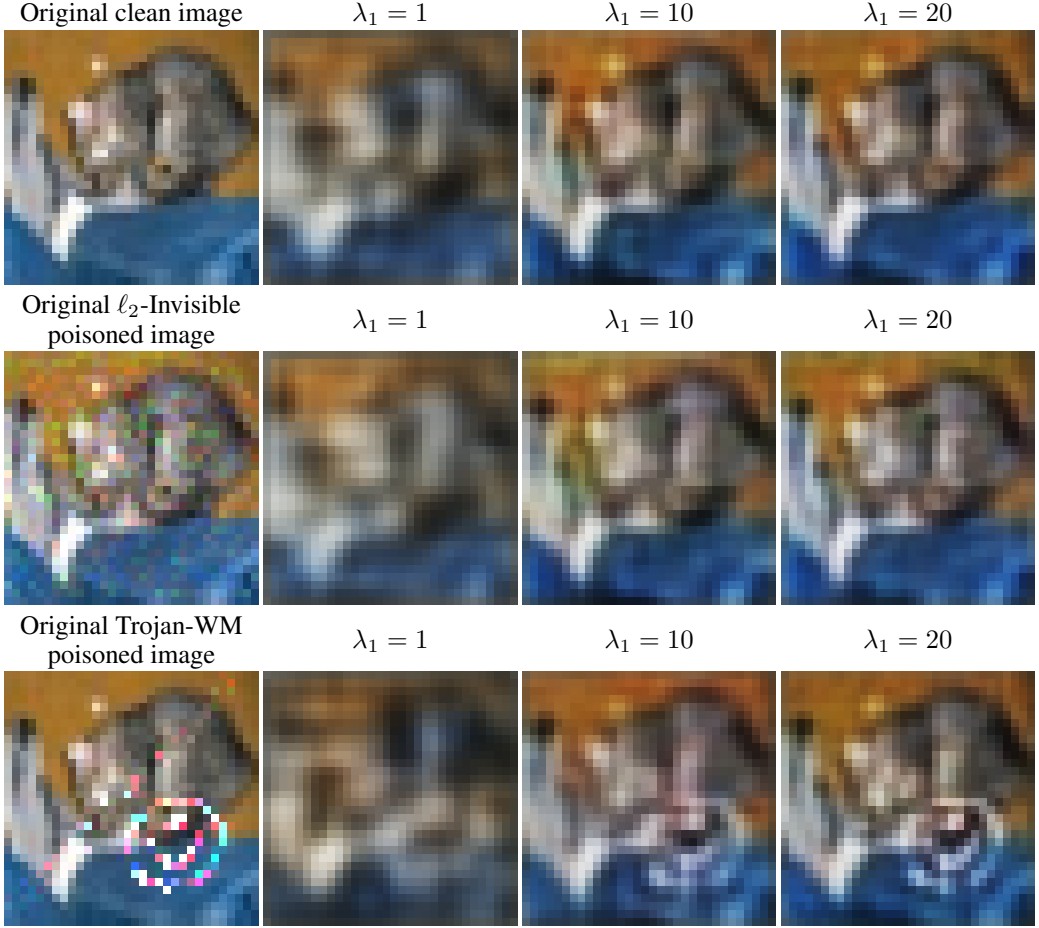

Figure 4: Qualitative results for image reconstruction in our T&R method on CIFAR10. The first row: The original clean image and its reconstructed versions under different $\lambda_1$ values. The second row: The original $\ell_2$-Invisible poisoned image and its reconstructed versions under different $\lambda_1$ values. The third row: The original Trojan-WM poisoned image and its reconstructed versions under different $\lambda_1$ values.

methods [44, 45, 46, 47] which can indirectly tell whether the training set is poisoned[7], and then decide whether it is necessary to apply backdoor defense methods. With that said, for completeness of the experiments, we show how the defense methods perform on clean training sets (i.e., datasets with poison ratio $\alpha = 0$).

We compare our method with the two strongest baselines, ABL and I-BAU, on the clean CIFAR10 training set without any backdoor attacks. The results are shown in Table 10. As we can see, compared with normal training, both I-BAU and our method can keep acceptable CA when trained on clean dataset. In contrast, ABL suffers considerable performance drop.

Although the original ABL paper reported no significant CA drop when ABL is applied on clean training sets [12], we believe this is because the authors of [12] used different hyper-parameter settings for ABL on clean and poisoned datasets. The results reported in Table 10 are obtained using the default hyper-parameters provided in the original ABL paper (i.e., the ones for best performance on poisoned datasets). Since we are assuming the defender has no prior knowledge on whether the dataset is poisoned or not, it is more reasonable to use the same hyper-parameters for both situations. In fact, it is quite intuitive to explain why ABL brings performance drop on clean training set. ABL selects a portion of training samples that is determined as potential backdoor training samples, and

---

[7]If a model trained from scratch on the dataset is detected as poisoned, then the training set is likely to be poisoned. Otherwise, it is likely to be clean.

then unlearns those selected samples. If there is no backdoor samples in the training set, then the selected samples are all clean ones, and unlearning on them will hurt clean accuracy.

Table 10: Clean accuracy (CA) of different defense methods on clean CIFAR10 training set without backdoor attacks. Note that it is not reasonable to compare the ASR here, since no backdoor is inserted into the model when the training set is clean.

| Normal training | ABL | I-BAU | Ours |
|---|---|---|---|
| 89.81 | 72.72 | 85.44 | 83.87 |

## C.5 Results of DP

Due to the space limit of the main text, we report the results of DP [30], which achieves the least competitive performance among the compared defense methods, in this section. As shown in Table 11 and Table 12, our method largely outperforms DP.

Table 11: Results of DP on CIFAR10.

| | DP | | Ours | |
|---|---|---|---|---|
| **Attack method** | **ASR** | **CA** | **ASR** | **CA** |
| BadNet-Grid | 53.21 | 27.73 | 1.21 | 84.42 |
| BadNet-White | 42.92 | 26.52 | 3.14 | 83.96 |
| Blend | 86.68 | 26.52 | 10.59 | 83.82 |
| $\ell_0$-Invisible | 35.90 | 25.00 | 2.91 | 84.04 |
| $\ell_2$-Invisible | 60.90 | 20.76 | 0.74 | 84.01 |
| Smooth | 95.41 | 28.18 | 4.23 | 83.63 |
| Trojan-SQ | 43.57 | 25.69 | 6.54 | 79.92 |
| Trojan-WM | 99.62 | 23.00 | 12.66 | 79.97 |
| SIG | 97.02 | 28.73 | 0.02 | 82.97 |
| LCBA | 82.80 | 18.95 | 5.41 | 82.57 |
| Average | 69.80 | 25.11 | 4.75 | 82.93 |

Table 12: Results of DP on GTSRB.

| | DP | | Ours | |
|---|---|---|---|---|
| **Attack method** | **ASR** | **CA** | **ASR** | **CA** |
| BadNet-Grid | 67.84 | 14.02 | 0.20 | 95.94 |
| BadNet-White | 52.20 | 16.14 | 0.01 | 95.74 |
| Blend | 83.26 | 12.87 | 1.98 | 95.62 |
| $\ell_0$-Invisible | 84.68 | 13.44 | 1.32 | 95.87 |
| $\ell_2$-Invisible | 82.49 | 16.94 | 0.03 | 96.10 |
| Smooth | 96.11 | 15.27 | 0.11 | 96.12 |
| Trojan SQ | 61.66 | 13.20 | 1.47 | 95.17 |
| Trojan WM | 99.11 | 13.19 | 7.06 | 91.95 |
| SIG | 94.71 | 18.92 | 0.54 | 94.56 |
| Average | 80.23 | 14.89 | 1.41 | 95.23 |

## D Difference in application scenario with previous works

Our method has less flexible application scenario compared with previous backdoor defense methods. Specifically, there are three common scenarios for backdoor defense:

Scenario 1: The defender gets a pretrained model from an untrusted source (e.g., the Internet), which is potentially backdoored. The defender has a small clean holdout set to sanitize the backdoored model, but don't have access to the original poisoned dataset.

Scenario 2: The defender collects raw data from an untrusted source (e.g., uploaded by untrusted users or from the Internet), and then trains the model on her own using the collected dataset, which potentially contains backdoor samples. The defender has a small clean holdout set to sanitize the backdoored model, as in Scenario 1.

Scenario 3: The defender collects raw data from an untrusted source (e.g., uploaded by untrusted users or from the Internet), and then trains the model on her own using the collected dataset, which potentially contains backdoor samples. The defender does not have a small clean holdout set to sanitize the backdoored model. This is a harder version than Scenario 2, since it doesn't require the defender to have a small clean holdout set.

Previous methods FP [3], NAD [4], I-BAU [5] are applicable in Scenario 1 and 2. Previous methods ABL [12] and DP [30] are applicable in Scenario 2 and 3. Our method is applicable only in Scenario 2. In Scenario 2, where all methods are applicable, our method achieves the best performance, outperforming previous methods by a considerable margin.

Scenario 2 is very common in the real world, compared with Scenario 1: In many cases, the defenders would train the model on their own, instead of directly using the (potentially backdoored) models released by a third-party. For example, the defender may use a model with some specific model size or architecture adapted for their hardware (e.g., mobile devices) with unique requirements, which will not be met by the third-party model. Or maybe the defender has a large amount of (potentially poisoned) internal data, which can lead to better performance than the third-party models trained on a dataset which is smaller and has distributional shifts. Or maybe the defender has its own advanced techniques to train a model for the specific task, which can lead to better performance than the general training techniques available to the third-party model trainer.

Scenario 2 does have one more requirement than scenario 3: It requires a small clean holdout set. We think this is reasonable, since previous methods [3, 4, 5] also have this requirement (in both scenario 1 and 2). In practice, to make sure the model achieves good performance before its deployment, the defender usually need to collect some clean samples for evaluation purpose. A small clean holdout set can be separated from the clean validation set.