# OpenReview forum: "Trap and Replace: Defending Backdoor Attacks by Trapping Them into an Easy-to-Replace Subnetwork"
_NeurIPS.cc/2022/Conference — NeurIPS 2022 Accept_

### Official Review · Reviewer_goHi · 2022-07-10

**Rating:** 5
**Confidence:** 5
**Soundness:** 3 good
**Presentation:** 4 excellent
**Contribution:** 3 good

**Summary:**

The authors of this work proposed a defense method against backdoor attacks. The defense method consists of two stages. In the first stage, they trapped the backdoors in a subnetwork. In the second stage, they replace the poisoned subnetwork and retrain the network with clean samples. Consequently, this method outperforms previous state-of-the-art methods.

**Questions:**

1. I think the reconstruction feature is pretty different from the classification feature which may cause both tasks to be difficult to train in more complex datasets. Can you show some reconstruction results? Does the reconstruction network reconstruct the trigger as well?
2. It is confusing that in the SIG attack on ImageNet, the clean accuracy becomes higher in Table 3. Why your finetuning is so powerful in this case?
3. Is your framework only suitable for convolutional neural networks? How about vision transformers?
4. As you mentioned in the footnote on page 2, semi-supervised learning and self-supervised learning are not naive solutions, since they are vulnerable to backdoor attacks. Is your work aim to provide a defense mechanism in this scenario?
5. As a suggestion and a question, I think that the poisoned data is aim to fool the classifier hence finetuning will work in this case. In multi-task learning, you can try to create poisoned data that aims to fool any downstream tasks. As finetuning might work for each task, how about putting various purpose attacks at the same time?


**Limitations:**

Yes. The authors state that there are no potential negative societal impacts of this work.

**Strengths And Weaknesses:**

Strengths:

1. The writing of this paper is very clear and easy to follow.
2. The experimental results show that the method outperforms the other six baseline methods in various attack methods except in some cases (Blend, Trojan SQ, Trojan WM).
3. The ablation study clearly shows the significance of the two stages.

Weaknesses:

1. In my opinion, the threat model is pretty unrealistic (with some holdout clean samples) however you can follow this assumption from previous works.
2. The auxiliary image reconstruction task encourages the stem network to keep sufficient low-level visual features that are hard-to-learn but semantically correct, protecting the stem network from overfitting to the easy-to-learn but semantically incorrect backdoor correlations. I cannot figure out the relation between this intuition and how the effect of poisoned data can be trapped.

Typos:
1. Line 156 has two “assumptions”.
2. Line 238 “Tojan” --> “Trojan”.

---

> ### Author Response · Authors · 2022-08-02
> **Response to Reviewer goHi**
>
> Thank you for your insightful comments and questions.
>
> Q1. The assumption that a small clean holdout set is available to the defender is not practical.
>
> As you have pointed out, this is a common assumption used by our and many previous methods [3,4,5]. We agree that this assumption can cause practical limitations of our method. However, we still would like to humbly defense for this assumption.
>
> To make sure the model achieves good performance before its deployment, the owner of the model (e.g., a company) usually need to collect some clean samples for evaluation purpose. A small clean holdout set can be separated from the clean validation set. Some practical ways to obtain those clean samples include using the company’s own trusted internal data or buying from trusted third parties.
>
>
> Q2. More explanations on the intuition.
>
> The image reconstruction task in stage 1 protects the stem network from overfitting to the easy-to-learn but semantically incorrect backdoor correlations. In contrast, no defense mechanisms are applied on the light-weighted classification head in stage 1. As a result, the classification head is more vulnerable than the stem network and more prone to learn the backdoor correlation. In other words, we “bait” the backdoor attack to the more vulnerable classification head.
>
> We also added a new Figure 4 in appendix to visualize the feature scatters learned with and without the image reconstruction head. It provides more intuitive explanations on why the stem network is protected from learning the backdoor correlations. Please check the updated appendix file for detailed results.
>
> Q3. Can you show some reconstruction results? Does the reconstruction network reconstruct the trigger as well?
>
> We added new visualization results in Figure 3 in Appendix D. Please check the updated appendix file for detailed results.
>
> The answer to the second question varies depending on the attack method. For example, for $\ell_2$-Invisible attack, the trigger is blurred out and totally unrecognizable in the reconstructed images. However, for Trojan-WM attack, the reconstructed images still keep a vague pattern of the trigger.
>
> Note that whether the reconstructed images keep the visual pattern of the backdoor trigger **has nothing to do with** the effectiveness of our defense method. It doesn’t matter whether the output features of the stem network encode the visual features of the backdoor trigger, as long as the **correlation** between the backdoor trigger and the target class is not learned. The newly added feature scatters in Figure 4 indicates that our method successfully prevents the stem network from learning the backdoor correlation.
>
> Q4. Why our method has higher clean accuracy than standard training on ImageNet-12 under SIG attack?
>
> Standard training on the clean ImageNet-12: clean accuracy=79.02%.
>
> Standard training on SIG poisoned ImageNet-12: clean accuracy=71.67%.
>
> Our method on SIG poisoned ImageNet-12: clean accuracy=77.33%.
>
> SIG attack on ImageNet degrades the image quality, making the clean accuracy drop by a considerable margin compared with training on clean ImageNet-12. Our method is finetuned on a clean holdout set, which may explain its superior clean accuracy than standard training on SIG poisoned ImageNet-12.
>
> Q5. Can our method generalize to ViT?
>
> We agree this is an important question. Due to time limit of the rebuttal phase, we can’t get conclusions before the deadline. We will continue investigating in this problem and hopefully show results in the final version.
>
> Q6.  Does this work aim to provide a defense mechanism in the self-supervised or semi-supervised scenario?
>
> Our method mainly focuses on the supervised learning scenario. We are also interested in how to generalize our method to the self-supervised or semi-supervised setting, which will be our future work.
>
> Q7. Multitask learning results.
>
> Following your suggestion, we use our method to defense backdoor attack in multitask learning. Specifically, we use two separate classification heads for CIFAR10 and GTSRB classification, and one image reconstruction head. All three head networks share the same stem network. $\ell_2$-Invisible attack is added on both CIFAR10 and GTSRB training set. The model is jointly trained on the union of the poisoned CIFAR10 and poisoned GTSRB datasets. The results are listed below:
>
> No defense:
>
> CIFAR10 head: ASR=100%, ACC=87.04%;
>
> GTSRB head: ASR=100%, ACC=94.91%.
>
> Our method:
>
> CIFAR10 head: ASR=1.80%, ACC=82.41%;
>
> GTSRB head: ASR=0.05%, ACC=93.80%.
>
> Our method can simultaneously defend multiple backdoor attacks in this multitask learning setting.

---

### Official Review · Reviewer_CZwS · 2022-07-11

**Rating:** 4
**Confidence:** 4
**Soundness:** 2 fair
**Presentation:** 3 good
**Contribution:** 2 fair

**Summary:**

The authors propose to trap the backdoor by a lightweight classification head on top of a low level feature extractor and replace it with a clean classifier to remove the backdoor. Extensive experimental results on different dataset against various attacks show the effectiveness of the proposed method.

**Questions:**

a. I am curious about the performance of the other methods. For example, in [11], ANP use 500 images (i.e., 1% of the training data) to purify the model against SIG on CIFAR-10, and the degradation of the clean accuracy is only 0.24% (93.64% to 93.40%). In this paper, the authors use 5% of the training data (2500 images), but the performance of ANP is far worse (89.27% to 85.67%). It is strange that the baseline methods perform such badly.

**Ethics Review Area:**

["Privacy and Security (e.g., consent)"]

**Limitations:**

No. I think the largest limitation is the threat model. As far as I know, this is the first method that require both a poisoned dataset and a clean holdout set. If so, I suggest the author to discuss the practical scenario of such threat model.

**Strengths And Weaknesses:**

Strengths:

a. The idea is interesting, and the motivation is reasonable.

b. The authors provide rich ablation experiments to evaluate the proposed method.

Weaknesses:

a. Most of the previous methods only need access to either the poisoned dataset or a small clean dataset, but the proposed method requires access to both, which limits its practical value.

b. According to Table 1, the degradation on clean accuracy is considerably large. This may because the final classifier is only trained on limited data (the holdout dataset).

---

> ### Author Response · Authors · 2022-08-02
> **Response to Reviewer CZwS**
>
> Thank you for your careful reading and insightful comments. We carefully address your concerns below and hope they can make you consider our work more favorably.
>
> Q1. “Most of the previous methods only need access to either the poisoned dataset or a small clean dataset, but the proposed method requires access to both, which limits its practical value.”
>
> We understand your concern that only practical assumptions should be used when designing methods. However, we humbly disagree with your claim that our method uses more assumptions than most previous backdoor defense methods. Please allow us to explain in more details bellow.
>
> On one hand, the existence of the poisoned dataset is the presupposition for all backdoor defense methods. There has to be a poisoned dataset at the very first place before we need to conduct any backdoor defense. On the other hand, many previous defense methods [3,4,5] also require a small clean holdout set, just like in our method.
>
> For example, in Algorithm 1 of the I-BAU paper [5], the poisoned model and the clean holdout set are both listed as the inputs. The poisoned model is obtained by training on the poisoned dataset. The same requirements are also used in FP [3] and NAD [4]. In other words, our method follows the assumptions used in [3,4,5], where two input datasets are required: a large poisoned training set (or equivalently a pretrained poisoned model in [3,4,5]) and a small clean holdout set.
>
> The difference between our method and [3,4,5] is not in which datasets are required, but in when the defense takes place. [3,4,5] use a post-hoc defense strategy: They first train the model on the poisoned dataset without any defense mechanism. The defense is only conducted after that first stage as a post-hoc sanitization process. However, their limitation is that once the backdoor features are learned in the first stage, it is hard to be unlearned in the second stage. Our method solves this problem by applying the defense mechanism at the very beginning of the training process (i.e., in the first training stage). Specifically, our method baits and traps the backdoors in a small and easy-to-replace subnetwork (i.e., the 1st stage of our method), making it much easier to remove the learned backdoor correlations from the network (i.e., the 2nd stage of our method).
>
> Q2. According to Table 1, the degradation on clean accuracy is considerably large.
>
> Compared with previous defense methods, our method has much smaller degradation on clean accuracy. For example, in the last row of Table 1, our method has 3.14% higher average clean accuracy and 5.04% lower average attack success rate than I-BAU on CIFAR10.
>
> Q3.  Why we reported a larger clean accuracy drop in ANP on SIG attack than the original paper?
>
> Thanks for your careful reading. We used a stronger attack setting for SIG compared with the original ANP paper. Specifically, we poison 100% of the target-class training samples when using SIG attack in our paper. In contrast, the ANP paper poisoned only 80% of the target-class training sample. In other words, the poisoning ratio of SIG attack in our paper is higher than that used in the ANP paper. Using our stronger attack setting, SIG achieves 99.93% attack success rate (ASR) on CIFAR10 when no defense method is applied (in the 1st column and the 3rd to the last row in Table 1). In contrast, the weaker attack setting used in ANP paper leads to a lower 94.26% ASR (numbers cited from the Appendix A of the ANP paper). The backdoor injected by stronger attacks is harder to get unlearned. This explains why ANP suffers higher clean accuracy drop when applied under the stronger attack setting used in our paper.

---

> > ### Comment · Reviewer_CZwS · 2022-08-03
> > **Reply to the Authors**
> >
> > Thanks for your detailed comments. The response has addressed part of my concerns. I appreciate the authors' work, which is intuitively correct and well performed. However, my concern about the threat model still remains.
> >
> > Firstly, it is right that the existence of the poisoned dataset is the presupposition for all backdoor defense methods (against poisoning-based backdoor attack), but the poisoned dataset may not be accessible by the defender. One of the typical scenarios is that when the poisoned models are directly provided by the adversary, and no poisoned datasets are available. The mentioned methods [3,4,5] (FP, NAD, I-BAU) do require poisoned models, but not poisoned datasets. The main difference is that FP, NAD and I-BAU can still be applied when poisoned datasets are not accessible, while the proposed T&R cannot. The author of [5] also mentioned that their method does not require the poisoned data: "Note that DP requires access to the poisoned data; hence, its attack model is different from the attack model of the other baselines and our method." (Line 2~3 in page 7). This led to an embarrassing situation: when compared with FP, NAD and I-BAU etc., T&R requires additional poisoned data, when compared with methods that only require poisoned data like ABL [12], DP [35], T&P need to access an extra clean holdout dataset. From this perspective, the comparison with other methods in the paper might not be fair.

---

> > > ### Author Response · Authors · 2022-08-03
> > > **Response to Reviewer CZwS**
> > >
> > > Thank you for your response. We agree with you that the application scenario of our method is less flexible than that of [3,4,5]. This is one limitation of our method. We will clearly discuss this limitation of our method in the final version. We sincerely thank you for pointing this out!
> > >
> > > There are three application scenarios:
> > >
> > > Scenario 1: The company gets a pretrained model from an untrusted source (e.g., the Internet), which is potentially backdoored. The company has a small clean holdout set to sanitize the backdoored model, but don't have access to the original poisoned dataset.
> > >
> > > Scenario 2: The company collects raw data from an untrusted source (e.g., uploaded by untrusted users or from the Internet), and then trains the model on her own using the collected dataset, which potentially contains backdoor samples. The company has a small clean holdout set to sanitize the backdoored model, as in Scenario 1.
> > >
> > > Scenario 3: The company collects raw data from an untrusted source (e.g., uploaded by untrusted users or from the Internet), and then trains the model on her own using the collected dataset, which potentially contains backdoor samples. The company does not have a small clean holdout set to sanitize the backdoored model. This is a harder version than Scenario 2, since it doesn't require the company to have a small clean holdout set.
> > >
> > > Previous methods FP [3], NAD [4], I-BAU [5] are applicable in Scenario 1 & 2.
> > >
> > > Our method is applicable in Scenario 2.
> > >
> > > Previous methods ABL [12] and DP [35] are applicable in Scenario 2 & 3.
> > >
> > > In Scenario 2, all methods are applicable, but our method achieves the best performance, outperforming all previous methods by a considerable margin.
> > >
> > > Scenario 2 is **very common** in the real world, compared with Scenario 1: In many cases, the companies would train the model on their own, instead of directly using the (potentially backdoored) models released by a third-party. For example, the company may use a model with some specific model size or architecture adapted for their hardware (e.g., mobile devices) with unique requirements, which won't be met by the third-party model. Or maybe the company has a large amount of (potentially poisoned) internal data, which can lead to better performance than the third-party models trained on a dataset which is smaller and has distributional shifts. Or maybe the company has its own advanced techniques to train a model for the specific task, which can lead to better performance than the general training techniques available to the third-party model trainer.
> > >
> > > Scenario 2 does have one more requirement than scenario 3: It requires a small clean holdout set. We think this is reasonable, since previous methods [3,4,5] also have this requirement (in both scenario 1 and 2). In practice, to make sure the model achieves good performance before its deployment, the company usually need to collect some clean samples for evaluation purpose. A small clean holdout set can be separated from the clean validation set.
> > >
> > > In summary:
> > >
> > > 1) We agree the application scenario of our method is less flexible than previous methods. We will clearly discuss this limitation in our final version. We sincerely thank the reviewer for pointing this out.
> > >
> > > 2) Although relatively more limited than those of previous methods, the application scenario of our method is still very common and practical in the real world.
> > >
> > > 3) In the application scenario where our method is applicable, our method achieves considerably better performance than previous works.
> > >
> > > We hope our explanation can make you consider our work more favorably.
> > >
> > > Thank you!

---

> > > > ### Comment · Reviewer_CZwS · 2022-08-06
> > > > **Response to the response**
> > > >
> > > > Sorry for the late reply. I really appreciate the authors detailed comments. I think the discussion about the practical scenario is one of the most important parts of this work, as it uses a different setting as the previous defense methods. Here are my thoughts about the mentioned scenario 2:
> > > >
> > > > I'm trying to think about the more specific settings in scenario 2. In my opinion, there are three different cases:
> > > >
> > > > __1)__ The company collects untargeted data to try to train a pretrained model for the downstream target task. And the holdout clean dataset will be used for fine-tuning. In this case, the training pipeline is a typical transfer learning, where the source data has different distribution (including input distribution and label distribution) as the target distribution. Note that in this case it is much harder to perform targeted attack.
> > > >
> > > > The company collects data with specific labels which is the same as the target task. This can be either __2)__ a transfer learning task (domain adaptation, where the input distribution is different between the source and target task) or __3)__ a vanilla i.i.d. training task (the company filters the untrusted dataset to construct the clean holdout dataset).
> > > >
> > > > In summary, most of tasks in scenario 2 are about transfer learning, which is not discussed in this paper. The experimental results in the paper only show that T&R performs well in 3). These are my preliminary thoughts, and the authors are welcome to pointed out anything that may missed.
> > > >
> > > > I will keep my score for now, but it doesn't mean that I don't like the authors' method. Because I still think the practical application may be limited. But if the other reviewers think it doesn't matter, I'm fine with it.

---

> > > > > ### Author Response · Authors · 2022-08-07
> > > > > **Response to Reviewer CZwS**
> > > > >
> > > > > Thank you for your detailed comments and insightful opinions! We think these constructive discussions are beneficial, to not only the authors, but also the entire community.
> > > > >
> > > > > We agree that our experiments are conducted for the I.I.D. setting (i.e., "case 3") you mentioned. Below we would like to humbly defense for the popularity of "case 3".
> > > > >
> > > > > As mentioned in the label-consistent backdoor attack (LCBA) paper [20], "one particular vulnerability stems from the fact that state-of-the-art ML models are trained on large datasets, which, unfortunately, are expensive to collect and curate. It is thus common practice to use training examples sourced from *a variety of, often untrusted, sources*."
> > > > >
> > > > > Our assumption is: Amount all the data sources, there are a small portion of well-trusted ones, which have largely the same data distribution as the untrusted sources.
> > > > >
> > > > > For example, an autonomous driving company employs multiple agents to collect self-driving video data. The data collected by different agents are largely from the same distribution since they use the same hardware configurations (e.g., the same type of autonomous vehicle with the same type of camera) provided (and required) by the company. On one hand, there may be some malicious data collection agents (i.e., the attackers) trying to inject backdoor samples to the collected training data. On the other hand, there are also a small portion of well-trusted agents proving the small clean holdout set.
> > > > >
> > > > > Note that previous works [3,4,5] also used similar assumptions: The provider of the clean holdout set is well-trusted and not cooperating with the attacker, and that the clean holdout set (e.g., clean CIFAR10 images) has the same distribution of the poisoned training set (e.g., poisoned CIFAR10 training set).
> > > > >
> > > > > Other common I.I.D. scenarios can be found in medical image analysis. For example, when a medical institution or company collects chest X-rays to train a computer-aided diagnostic (CAD) model for tuberculosis diagnostic, samples collected by different (either untrusted or well-trusted) agents are largely from the same distribution since they are collected using the same type of X-ray machine required and provided by the company.
> > > > >
> > > > > The requirements and available resources for different practical problems can be diverse. Thus, we believe it important to provide solutions for different practical scenarios.
> > > > >
> > > > > We agree that it is important to make it clear to the readers about the problem settings used by different methods, so that they can make the best choice for their own applications. We will clearly describe the above-mentioned application scenarios in the final version. We also agree the transfer learning setting you mentioned is an important future work, which we will point out in the final version.
> > > > >
> > > > > We would like to thank you again for appreciating our technical method, and we hope our response can solve your concern on the limitations of the application scenario.
> > > > >
> > > > > Thank you!
> > > > >
> > > > > [20] Alexander Turner, Dimitris Tsipras, and Aleksander Madry. Label-consistent backdoor attacks.

---

> > > ### Author Response · Authors · 2022-08-06
> > > **Follow-up discussions**
> > >
> > > Dear Reviewer CZwS,
> > >
> > > Thank you for reviewing our paper. We have tried to answer your insightful questions carefully. We would appreciate it if you could share your thoughts on it.
> > >
> > > Thank you!

---

### Official Review · Reviewer_Nfpj · 2022-07-12

**Rating:** 7
**Confidence:** 4
**Soundness:** 4 excellent
**Presentation:** 3 good
**Contribution:** 3 good

**Summary:**

This paper proposes a new defense strategy *Trap and Replace* to protect deep neural networks from the backdoor in the poisoned dataset. This paper presumes that the backdoor pattern is easy to learn, so it trains a standard image classification model consisting of a stem network and a classification head and an extra image reconstruction model, which consists of the stem network and reconstruction head at the same time to encourage stem model to learn the low-level visual features. Then freeze the stem work and initialize the classification head by setting those parameters to random values. And train the new model in a small but clean dataset. The experimental results show that Trap and Replace outperforms SOTA defense strategies in most cases.


**Questions:**

N/A

**Limitations:**

Comments:
1. In previous works, the classification head is considered the fully connected layer. You can explain the definition of classification head and reconstruction head in the introduction session.
2. In line 156, there is a repetitive word "assumptions".

**Strengths And Weaknesses:**

Strength:
1. Introducing a reconstruction task to force the stem model to learn visual features is a novel idea. It is self-supervised training that does not need extra labeled samples.
2. Training classification head-on clean dataset can mitigate the attack success rate and keep accuracy at a high level.
3. The paper's summary of relative works is well organized and comprehensive.
4. The experiments are well designed and clearly shown in the tables.
5. Ablation study showing that both Trap and Replace are necessary is convincing.


Weakness:
1. Why using the fully connected layer as classification head fails to defend the backdoor should be discussed.
2. How many layers should be chosen as the classification head in others models that are not included in this paper should be considered.

---

> ### Author Response · Authors · 2022-08-02
> **Response to Reviewer Nfpj**
>
> Thank you for appreciating our work. We are glad to respond to your constructive questions and comments.
>
> Q1. Why using the fully connected layer as classification head fails to defend the backdoor (as shown in the first row in Table 5)?
>
> Intuitively, it is easier to trap the backdoor attacks in a larger subnetwork (e.g., a larger classification head). If the classification head $f_c$ has only one fully connected layer, then it is easy for the backdoor to scape to shallower layers.
>
> Q2. How many layers should be chosen as the classification head in different model architectures?
>
> In Table 5, we showed that the best results are achieved when the classification head is the last three layers (one fully connected layer and two convolutional layers) in WRN16-1. Following your suggestion, we conducted the same ablation study on a different model architecture – WRN28-2. The results on CIFAR10 with $\ell_2$-Invisible attack are listed below:
>
> Number of layers in $f_c$=1: ASR=100%, CA=92.03%;
>
> Number of layers in $f_c$=3: ASR=1.25%, CA=89.16%;
>
> Number of layers in $f_c$=5: ASR=0.74%, CA=75.08%.
>
> We suggest using 3-layer classification head in this case for the best trade-off between ASR and CA.
>
> Q3. Definition of reconstruction head.
>
> Thank you for your suggestion. We will add its definition in the introduction in the final version.

---

### Official Review · Reviewer_cjmb · 2022-07-21

**Rating:** 6
**Confidence:** 3
**Soundness:** 3 good
**Presentation:** 3 good
**Contribution:** 3 good

**Summary:**

The proposed work introduces a defense which uses an auxiliary reconstruction task along with the classification to ‘trap’ the backdoor within the classification head. In the second stage, a new classification head is trained from scratch to completely remove the effect of the backdoor. The intuition is that the auxiliary task ensures that low-level features of the image is preserved within the backbone or the stem network, hence reducing the effectiveness of the backdoor. Results are shown on CIFAR-10, GTSRB and ImageNet-12 against various attacks.

**Questions:**

— It seems that replacing the classification head without the auxiliary task results in significant drop in ASR. Is it possible to consider an experiment where the learning rate for samples from clean holdout set is 10x the learning rate for the untrusted dataset? There is no auxiliary task in this scenario. Such an experiment would show whether the effectiveness of the defense is purely due to the correct decision boundaries learned by the classification head or if it is due to the trapping mechanism.

— A more detailed ablation study on the size of clean holdout set is necessary. From, Table 9 of the appendix, it seems there is no difference between the 2.5% and 5% setting. It would be interesting to know what is the least amount of samples required to achieve similar results.


After Rebuttal:
Thanks to the authors for providing the rebuttal. Having gone through the other reviews, I will keep my score unchanged and strongly encourage the authors to include some of the new experiments about the relationship between reconstruction and ASR in the main paper.

**Limitations:**

Yes

**Strengths And Weaknesses:**

**Strengths:**

— The defense method introduced is novel and effective. It is also easy to use without requiring too much hyper-parameter tuning compared to previous methods.

— Authors conduct extensive ablation study to illustrate the effectiveness of the method. An Adaptive attack is also considered, against which the defense still remains effective.

— Results are shown on a variety of backdoor attacks, indicating that it can be used to defend against multiple attacks.

**Weaknesses:**

— Although the authors do provide an intuition that low-level features are preserved better with the auxiliary task, it is not clear why the proposed method should be effective. A feature analysis on the difference with and without the reconstruction task can be helpful in identifying why the defense is effective to such a degree.

— The reconstruction task proposed by the authors is not very well understood. An experiment where the quality of reconstructed images vs ASR can improve understanding. Although authors do provide a variant of this experiment in Table 8 of the appendix, a qualitative analysis would be better in this scenario. Another experiment would be to vary the size or capacity of the ‘decoder’ and observe the variation in ASR.

— Similar to previous methods, a clean heldout dataset is required which is typically 10%. This can be difficult to obtain for larger datasets.

---

> ### Author Response · Authors · 2022-08-02
> **Response to Reviewer cjmb**
>
> Thank you for your careful reading and acknowledging the novelty and effectiveness of our method. We are glad to answer your insightful questions.
>
> Q1. Analyses the difference between features learned with and without the reconstruction head.
>
> Following your suggestion, we added new visualization results in Figure 4 in Appendix D. In summary, the visualization results show a two-fold conclusion. First, when trained without the image reconstruction task, the stem network ignores the semantic features and overfits to the easy-to-learn but semantically incorrect backdoor correlations on backdoored samples. Second, when trained with the image reconstruction task, the stem network successfully preserves the semantically correct features on the poisoned samples. Please check the updated appendix file for detailed results.
>
> Q2. What is the relation between image reconstruction quality and backdoor attack success rate (ASR)?
>
> To investigate their relation, we design two sets of experiments.
>
> 2.1. The first is to vary the value of $\lambda_1$ in Eq (1). The larger $\lambda_1$, the better reconstruction results. As you have noticed, this is what we did in Table 8 in appendix. Following your suggestion, we added qualitative results of image reconstruction under different $\lambda_1$ value in Figure 3 in appendix. We also added another row in Table 8 to show the mean square error (MSE) of image reconstruction. Please check the updated appendix file for detailed results.
>
> Same conclusions can be drawn from the new visualization results as those from the original Table 8: A proper amount of image reconstruction quality is required to get good performance. On one hand, if $\lambda_1$ is too small (e.g., $\lambda_1$<1), then the supervision from the image reconstruction task is too weak, and thus the reconstruction quality is bad and ASR is high. On the other hand, if $\lambda_1$ is too large (e.g., $\lambda_1$=20), then the clean accuracy (CA) decreases because the stem network is biased towards learning image reconstruction features while ignoring the classification features.
>
> 2.2. Following your suggestion, we designed a second experiment by varying the capacity of the decoder (i.e., the image reconstruction head). Specifically, we use decoders with different channel numbers. For example, $1\times$ is the original decoder we used, and $2\times$ is the decoder with twice the channel numbers in all decoder layers (and thus with larger model capacity). The results are listed below:
>
> $1\times$ channel-width decoder: ASR=0.74%, CA=84.01%, MSE=0.0085
>
> $2\times$ channel-width decoder: ASR=0.71%, CA=83.49%, MSE=0.0082
>
> As we can see, increasing the capacity of decoder doesn’t significantly benefit the reconstruction quality (in terms of MSE) or the ASR.
>
> Q3. Similar to previous methods, a clean holdout dataset is required, which is a limitation of the proposed method.
>
> We agree this is a limitation of our method, which also commonly exists in other backdoor defense methods such as [3, 4, 5] as you pointed out. Note that on CIFAR10 and GTSRB, the size of our holdout set is 5% instead of 10%.
>
> Q4. Setting the learning rate for samples from clean holdout set to 10x the learning rate for the untrusted dataset, with no image reconstruction task used.
>
> Thank you for suggesting this new experiment. The results on CIFAR10 dataset with l2-invisible attack are: ASR=98.44%, ACC=86.60%. In other words, it can’t defend backdoor attack, showing the necessity of our backdoor trapper (i.e., the image reconstruction task).
>
> Q5. More ablation study results on the size of clean holdout set.
>
> Following your suggestion, we have added another two columns in Table 9 in appendix, which show the results when the holdout set sizes are 1% and 0.5% (i.e., 500 and 250 images on CIFAR10), respectively. In summary, decreasing the size of clean holdout set only affects the final clean accuracy but not the attack success rate. This is very intuitive since the defense of the stem network is done in the first training stage of our method and has nothing to do with the holdout set. The size of the holdout set only affects the quality of the clean classification head learned in the second stage. Please check the updated appendix file for detailed results.

---

### Meta-Review · Area_Chair_pT8F · 2022-08-23

**Recommendation:** Accept
**Confidence:** Less certain

**Metareview:**

The recommendation is based on the reviewers' comments, the area chair's personal evaluation, and the post-rebuttal discussion.

This paper proposed a new training method to defend against backdoor attacks. While all reviewers see merits in this paper, some discussions about (1) the practicality of the defense using clean data samples and (2) fair comparisons to existing defenses have been raised and discussed. During the author-reviewer discussion phase, the reviewer had detailed interactions with the authors to clarify different use cases and practical scenarios of the proposed defense and the fairness of the evaluation. So both major concerns are adequately addressed. Another reviewer also champions acceptance in the internal discussion. All in all, I am recommending acceptance. My confidence is lower compared to other submissions simply because this paper has the lowest average rating score of all papers I recommend acceptance.

**Award:**

No

---

### Decision · Program_Chairs · 2022-09-14

Accept